# Tracing a protein's folding pathway over evolutionary time using ancestral sequence reconstruction and hydrogen exchange

Shion An Lim[1,2†], Eric Richard Bolin[2,3†], Susan Marqusee[1,2,4,5]*

[1]Department of Molecular and Cell Biology, University of California, Berkeley, Berkeley, United States; [2]Institute for Quantitative Biosciences, University of California, Berkeley, Berkeley, United States; [3]Biophysics Graduate Program, University of California, Berkeley, Berkeley, United States; [4]Department of Chemistry, University of California, Berkeley, Berkeley, United States; [5]Chan Zuckerberg Biohub, San Francisco, United States

**Abstract** The conformations populated during protein folding have been studied for decades; yet, their evolutionary importance remains largely unexplored. Ancestral sequence reconstruction allows access to proteins across evolutionary time, and new methods such as pulsed-labeling hydrogen exchange coupled with mass spectrometry allow determination of folding intermediate structures at near amino-acid resolution. Here, we combine these techniques to monitor the folding of the ribonuclease H family along the evolutionary lineages of *T. thermophilus* and *E. coli* RNase H. All homologs and ancestral proteins studied populate a similar folding intermediate despite being separated by billions of years of evolution. Even though this conformation is conserved, the pathway leading to it has diverged over evolutionary time, and rational mutations can alter this trajectory. Our results demonstrate that evolutionary processes can affect the energy landscape to preserve or alter specific features of a protein's folding pathway.

DOI: https://doi.org/10.7554/eLife.38369.001

*For correspondence:
marqusee@berkeley.edu

†These authors contributed equally to this work

Competing interests: The authors declare that no competing interests exist.

## Introduction

Protein folding, the process by which an unfolded polypeptide chain navigates its energy landscape to achieve its native structure (*Dill and MacCallum, 2012*; *Baldwin, 1975*), can be defined by the partially folded conformations (intermediates) populated during this process. Such intermediates are key features of the landscape; they can facilitate folding, but they can also lead to misfolding and aggregation, resulting in a breakdown of proteostasis and disease (*Karamanos et al., 2016*; *Chiti and Dobson, 2017*; *Ahn et al., 2016*). While identifying and characterizing these intermediates is critical to understanding and engineering a protein's energy landscape, their transient nature and low populations present experimental challenges. Currently, we know very little about how evolutionary variations in the primary amino acid sequence affect these transient conformations and the overall folding pathway of a protein. Recent technological improvements in hydrogen exchange monitored by mass spectrometry (HX-MS) have provided access to the structural and temporal details of folding intermediates at near-single amino-acid resolution (*Hu et al., 2013*; *Walters et al., 2012*; *Mayne et al., 2011*; *Aghera and Udgaonkar, 2017*; *Vahidi et al., 2013*; *Khanal et al., 2012*). This pulsed-labeling HX-MS approach is particularly well suited to studies of multiple variants or families of proteins, as it does not require large amounts of purified protein or NMR assignments, and is a powerful platform to test hypotheses on how folding pathways and protein conformations

are dictated by sequence and the environment. Thus, pulsed-labeling HX-MS can be used to address long-standing questions in the field: How robust is a protein's energy landscape to changes in the amino acid sequence, and how conserved is the folding trajectory over evolutionary time?

Ribonuclease HI (RNase H) is an ideal system to investigate protein folding over evolutionary time. RNase H from *E. coli*, ecRNH* (the asterisk denotes a cysteine-free variant of RNase H), is arguably one of the best-characterized proteins in terms of its folding pathway and energy landscape. Both stopped-flow ensemble studies and single-molecule optical trap experiments demonstrate that this protein populates a major obligate intermediate before the rate-limiting step in folding (*Raschke and Marqusee, 1997*; *Raschke et al., 1999*; *Cecconi et al., 2005*; *Rosen et al., 2014*). A rare population of this intermediate can also be detected under native-state conditions (*Chamberlain et al., 1996*). Several homologs of RNase H have also been studied, yielding insight into the folding trends of extant RNases H (*Hollien and Marqusee, 2002*; *Kern et al., 1998*; *Ratcliff et al., 2009*). Given the wealth of data available on several extant RNases H, this seemed like an ideal system to explore the evolutionary basis of these observations.

One can use a phylogenetic technique called ancestral sequence reconstruction (ASR) to access the evolutionary history of a protein family and study the properties of ancestral proteins (*Harms and Thornton, 2013*; *Wheeler et al., 2016*). ASR has been applied to a variety of protein families and in addition to revealing the evolutionary history, these ancestral proteins can act as intermediates in sequence space to uncover mechanisms underlying protein properties (*Starr et al., 2017*; *Gaucher et al., 2008*; *Hobbs et al., 2012*; *Perez-Jimenez et al., 2011*; *Risso et al., 2013*; *Smock et al., 2016*; *Akanuma et al., 2013*; *Siddiq et al., 2017*). Recently, ancestral sequence reconstruction was applied to the RNase H family and the thermodynamic and kinetic properties of seven ancestral proteins connecting the lineages of *E. coli* and *T. thermophilus* RNase H (ecRNH* and ttRNH*) were characterized (*Hart et al., 2014*; *Lim et al., 2016*; *Lim and Marqusee, 2017*). Stopped-flow kinetics monitored by circular dichroism (CD) demonstrate that all seven ancestral proteins populate a folding intermediate before the rate-limiting step. Additionally, the folding and unfolding rates show notable trends along the phylogenetic lineages, and the presence of a folding intermediate plays an important role in modulating these evolutionary trends (*Lim et al., 2016*). Although the kinetic mechanism appears conserved, the structural similarities and differences between the folding intermediates across these RNases H are unknown.

For ecRNH*, multiple methods have confirmed the structural details of the folding intermediates. This major folding intermediate, termed $I_{core}$, which forms before the rate-limiting step, involves secondary structure formation in the core region of the protein, including Helices A-D and Strands 4 and 5, while the rest of the protein (Helix E and Strands 1, 2, 3), remains unfolded (*Figure 1A*) (*Raschke and Marqusee, 1997*; *Spudich et al., 2004*; *Connell et al., 2009*). Pulsed-labeling HX-MS with near amino acid resolution was developed using ecRNH* as the model protein (*Hu et al., 2013*). This approach confirmed the structure of $I_{core}$ and revealed the stepwise protection of individual helices leading up to the intermediate. Specifically, the amide hydrogens in Helix A and Strand four are the first elements to gain protection, followed by those in Helix D and Strand 5, and then Helices B and C to form the canonical $I_{core}$ intermediate. The periphery, comprising of Strands 1–3 and Helix E, gains protection in the rate-limiting step to the native state. Is this $I_{core}$ folding intermediate and the stepwise folding pathway conserved across evolution?

Here, we used pulsed-labeling HX-MS on the resurrected family of RNases H to investigate the evolutionary and sequence determinants governing the folding trajectory. Specifically, we find that the structure of the major folding intermediate ($I_{core}$) has been conserved over three billion years of evolution, suggesting that this partially folded state plays a crucial role in the folding or function of the protein. However, the path to this intermediate varies, both during evolution and by designed mutations. The very first step in folding differs between the two extant homologs: for ecRNH*, Helix A gains protection before Helix D, while for ttRNH*, Helix D acquires protection before Helix A. This pattern can be followed along the evolutionary lineages: most of the ancestors fold like ttRNH* (Helix D before Helix A) and a switch to fold like ecRNH* (Helix A before Helix D) occurs late along the mesophilic lineage. These phylogenetic trends allow us to investigate how these early folding events are encoded in the amino acid sequence. By using single-point mutations to selectively modulate biophysical properties, notably intrinsic helicity of specific secondary structure elements, we are able to favor or disfavor the formation of specific conformations during folding and have engineering control over the folding pathway of RNase H.

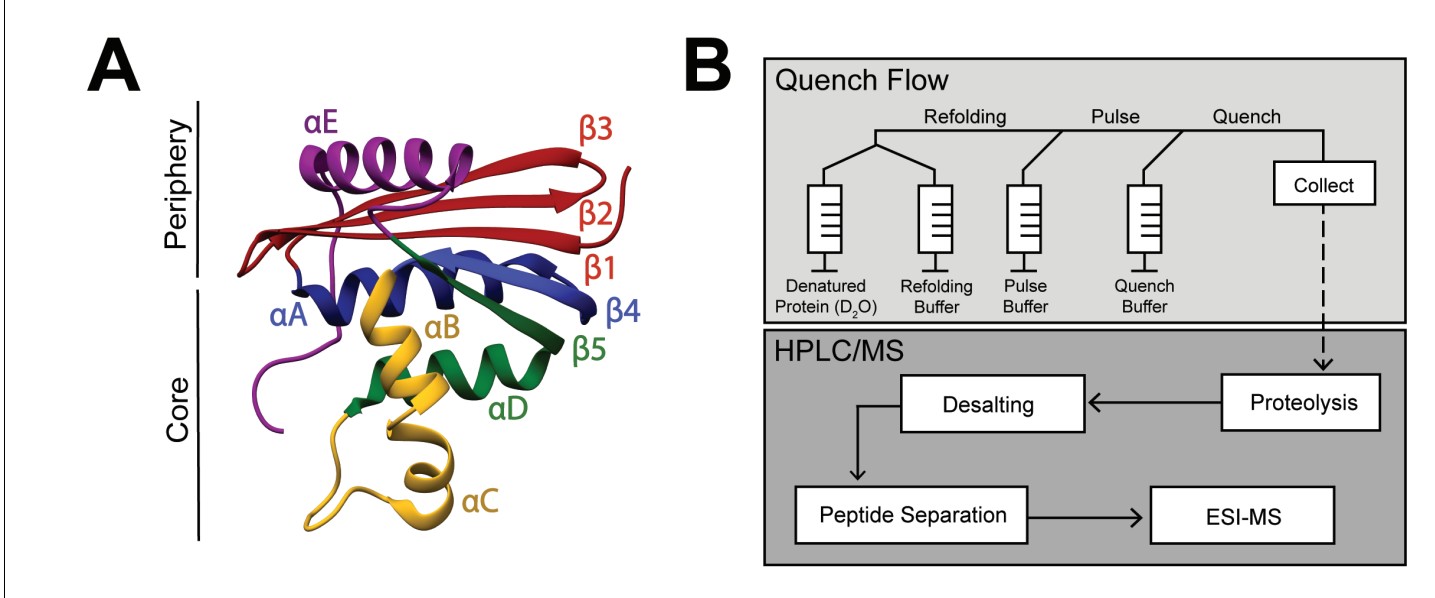

**Figure 1.** RNase H structure and schematic of pulsed-labeling HX-MS experiment (A) Crystal structure of *E. coli* RNase H* (ecRNH*) (PDB: 2RN2) (*Katayanagi et al., 1992*). Secondary structural elements: Red: Strand 1, Strand 2, Strand 3; Blue: Helix A, Strand 4; Yellow: Helix B, Helix C; Green: Helix D, Strand 5; Purple: Helix E. The core region of the protein ($I_{core}$) involving Helix A, Strand 4, Helix B, Helix C, Helix D, Strand 5 and the periphery region of the protein involving Strand 1, Strand 2, Strand 3, Helix E are denoted. (B) Pulsed-labeling setup and workflow. Unfolded, fully deuterated protein in high [urea] is rapidly mixed with low [urea] refolding buffer to initiate refolding. After some refolding time, hydrogen exchange of unprotected amides is initiated by mixing with high-pH pulse buffer. The hydrogen exchange reaction is quenched by mixing with a low-pH quench buffer. The sample is injected onto an LC-MS for in-line proteolysis, desalting, and peptide separation by reverse-phase chromatography followed by MS analysis.

DOI: https://doi.org/10.7554/eLife.38369.002

## Results

### Monitoring a protein's folding trajectory by pulsed-labeling HX-MS

We used pulsed-labeling hydrogen exchange monitored by mass spectrometry (HX-MS) on extant, ancestral, and site-directed variants of RNase H to examine the robustness of a protein's folding pathway to sequence changes. These experiments allow us to characterize the partially folded intermediates and the order of structure formation during folding to ask whether these intermediates have changed over evolutionary time, and what role sequence might play in determining these intermediate conformations.

*Figure 1B* outlines the scheme for the pulsed-labeling experiment (for details, see Materials and methods). Briefly, folding is initiated by rapidly diluting an unfolded (high [urea]), fully deuterated protein into folding conditions (low [urea]) at 10°C. After various folding times ($t_f$), a pulse of hydrogen exchange is applied to label amides in regions that have not yet folded. The amount of exchange at each folding timepoint is then detected by in-line proteolysis and LC/MS. Data are analyzed first at the peptide level by monitoring the protection of deuterons on peptides as a function of refolding time, and then at the residue level, using overlapping peptides de-convoluted by the program HDsite (*Kan et al., 2013*; *Kan et al., 2011*). Protection from exchange in these experiments arises from formation of structure, however it is impossible to tell the exact structure using hydrogen exchange alone. Thus, nonnative or transient structures can be detected, but the characterization of them is limited.

Since the original folding studies on many RNases H were carried out at 25°C, we re-characterized the folding of each RNase H variant at 10°C using stopped-flow circular dichroism spectroscopy (*Figure 2—figure supplement 1*, *Figure 2—source data 1*). The refolding profiles were consistent with those at 25°C (*Raschke and Marqusee, 1997*; *Hollien and Marqusee, 2002*; *Lim et al., 2016*). At low [urea], all ancestors show a large signal change (burst phase) within the dead time of the

stopped-flow instrument (~15 msec), followed by a slower observable phase which fit well to a single exponential. The resulting chevron plots (ln($k_{obs}$) vs [urea]) show the classic rollover at low [urea] due to the presence of a stable folding intermediate. As expected, the observed rates at 10°C are slower than 25°C, but the chevron profiles are similar for all RNase H variants. Thus the overall folding trajectory, notably the population of a folding intermediate, has not changed between the two temperatures.

## Monitoring the folding pathway of ttRNH* using pulsed-labeling HX-MS

First, we characterized the conformations populated during folding of extant RNase H from *T. thermophilus* and compared its folding trajectory to the previously characterized folding trajectory of *E. coli* RNase H (*Figure 2—figure supplement 1*, *Figure 2—figure supplement 2*, *Figure 2—source data 1*) (*Hu et al., 2013*). 374 unique peptides mapping to the ttRNH* sequence were identified by MS (*Figure 2A*, *Figure 2—source data 1*). The major folding intermediate in ttRNH*, $I_{core}$, is strikingly similar to that of ecRNH* (*Hu et al., 2013*). Similar to ecRNH*, peptides associated with $I_{core}$ (Helix A-D, Strands 4 – 5) gain protection early (within milliseconds), corresponding to the timescale for the formation of the folding intermediate. Peptides associated with the periphery of the protein (Strands 2 – 3, Helix E) gain protection on the order of seconds, corresponding to the rate-limiting step (*Figure 2B*). Although it has been shown by CD spectroscopy that the slow observed refolding kinetics of ttRNH* fit best to a biphasic exponential process (*Figure 2—figure supplement 1*, *Figure 2—source data 1*), we were not able to observe structural changes that can be attributed to these two phases by pulsed-labeling HX-MS. Thus, uncovering the molecular mechanism that gives rise to the biphasic behavior of ttRNH* remains an open question.

Looking at the very early refolding times allows one to determine the individual folding steps preceding $I_{core}$.. At the earliest time point (~1 msec), almost all peptides are unfolded (fully exchange with solvent) with the exception of those in Helix D and Strand 5, which are ~40% deuterated (*Figure 2C*). Peptides spanning Helix A and Strand 4 are less protected (~15% deuterated) at this same time point. This order of protection (Helix D before Helix A) is notably different than that for *E. coli* RNase H*, where Helix A is protected before Helix D (*Hu et al., 2013*). Peptides spanning Helix B and Helix C gain protection in the $I_{core}$ intermediate. Peptides from Strands 1 – 3 and Helix E do not gain full protection until significantly later (on the order of seconds), corresponding to the rate-limiting step to the native state. Thus, while the $I_{core}$ intermediate is largely conserved between ttRNH* and ecRNH*, the initial steps of folding differ between the two homologs.

The peptide data from each time point were also analyzed using HDSite to determine residue-level protection in a near site-resolved manner (*Figure 2D*). These site-resolved data also show protection appearing first in Helix D and Strand 5, followed by Helix A/Strand 4, Helix B/C, and finally, the periphery Helix E and Strands 1 – 3. The differences in the order of protection leading up to $I_{core}$ of ecRNH* and ttRNH* are also evident in this site-resolved analysis.

## Pulsed-labeling HX-MS on the ancestral RNases H

To look for evolutionary trends in the folding trajectory, we probed the folding pathway of ancestral RNases H along the lineages of *E. coli* and *T. thermophilus* RNase H (*Figure 3A*). Anc1* is the last common ancestor of ecRNH* and ttRNH*. Anc2* and Anc3* are ancestors along the thermophilic lineage leading to ttRNH*, and AncA*, AncB*, AncC*, and AncD* are ancestors along the mesophilic lineage leading to ecRNH*. Previous kinetic studies demonstrated that all of the ancestral proteins fold via a three-state pathway, populating an intermediate before the rate-limiting step (*Lim et al., 2016*; *Lim and Marqusee, 2017*). We now use pulsed-labeling HX-MS to obtain a near-site resolved trajectory of the folding pathway for each ancestor and determine whether the $I_{core}$ structure is conserved over evolution.

We obtained good peptide coverage for all of the ancestors with a minimum of 81 peptides seen in all time points for each variant (*Figure 3*, *Figure 3—figure supplements 1–6*, *Figure 3—source data 1*). As observed in both ttRNH* (above) and ecRNH* (*Figure 2*, *Figure 2—figure supplement 2*) (*Hu et al., 2013*), all of the ancestral RNases H populate the canonical $I_{core}$ folding intermediate prior to the rate-limiting step. Peptides corresponding to the $I_{core}$ region of the RNase H structure become protected on the timescale of milliseconds, while the rest of the protein gains protection on the timescale of seconds (*Figure 3C*, *Figure 3—figure supplements 1–6*). Thus, the structure of this

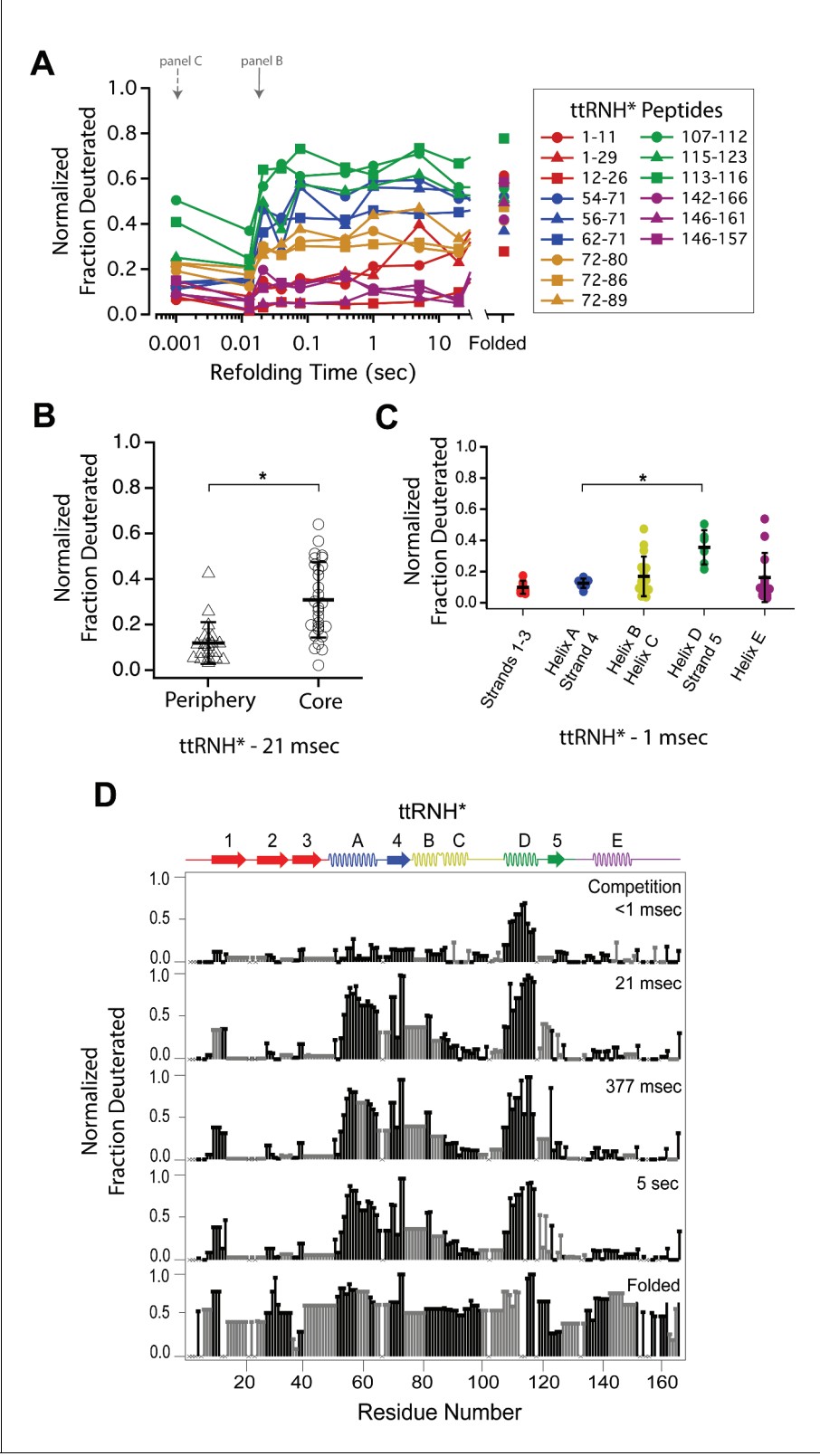

**Figure 2.** Determination of the folding pathway of *T. thermophilus* RNase H* by HX-MS. (**A**) Protection of representative peptides from ttRNH* at various refolding times. Peptides are colored according to their corresponding structural element. The solid arrow indicates the refolding time point analyzed in panel B. The dotted arrow indicates the refolding time point analyzed in panel C. (**B**) Protection of peptides mapping to the core region ($I_{core}$) or the periphery region of ttRNH* at 21 msec after refolding. Bars represent the mean and standard deviation of each data set. *p<0.0001

*Figure 2 continued on next page*

*Figure 2 continued*

(Welch's unpaired T-test) (C) Protection of peptides of ttRNH* mapping to distinct secondary structural elements at one msec after refolding. Bars represent the mean and standard deviation of each data set. *p=0.0027 (Welch's unpaired T-test). Data for B and C represent aggregate data from three separate experiments. (D) Residue-resolved folding pathway of ttRNH* at representative refolding time points. Data points in black indicate residues that are site-resolved. Data points in grey indicate residues in regions with less peptide coverage and are thus not site-resolved with the neighboring residues. Residues where site-resolved protection could not be determined due to insufficient peptide coverage is denoted with a 'x'.
DOI: https://doi.org/10.7554/eLife.38369.003

The following source data and figure supplements are available for figure 2:

**Source data 1.** HX-MS data for ttRNH* and Chevron Plot data of RNase H variants.
DOI: https://doi.org/10.7554/eLife.38369.006
**Figure supplement 1.** Chevron plot of RNase H variants studied at 10°C and 25°C.
DOI: https://doi.org/10.7554/eLife.38369.004
**Figure supplement 2.** Determination of the folding pathway of ecRNH* by HX-MS.
DOI: https://doi.org/10.7554/eLife.38369.005

major folding intermediate is not only present in both extant RNases H, but is conserved over nearly three billion years of evolutionary history.

Similarly to the extant proteins, the periphery of the ancestral proteins gains protection on a much slower timescale (*Figure 3C*, *Figure 3—figure supplements 1–6*). The details of protection in this region, however, vary somewhat across the ancestors. The periphery becomes fully protected by the last time point in all ancestral proteins except for AncB* (*Figure 3—figure supplement 4*). AncB* was previously characterized to be non-two-state with a notable population of the folding intermediate under equilibrium conditions, (*Lim et al., 2016*) and the lack of protection in the periphery in the folded state of AncB* is consistent with this observation. For Anc1* and Anc2*, there are also notable differences in the time course of protection for the terminal helix, Helix E. For these two proteins, the peptides spanning Helix E are decoupled from Strands 1 – 3 (which show protection on the same timescale as global folding) and do not gain protection even in the folded state of the protein (*Figure 3B*, *Figure 3D*, *Figure 3—figure supplement 1*), suggesting that Helix E is improperly docked or poorly structured in Anc1* and Anc2*. Indeed, Helix E is known to be labile in ecRNH*: a deletion variant of ecRNH* without this final helix forms a cooperatively folded protein, (*Goedken et al., 1997*) and recent single-molecule force spectroscopy of ecRNH* showed that Helix E can be pulled off the folded protein under low force while the remainder of the protein remains structured (manuscript in preparation). It appears that Helix E may be further destabilized in Anc1* and Anc2* such that it does not show protection in the native state.

## The early folding steps of RNase H change across evolutionary time

Since the order of events leading to $I_{core}$ differs between the extant homologs, we examined whether the ancestral RNases H spanning the lineages of these two homologs show any trends in their early folding steps. For each ancestor, we analyzed the fraction of deuterium protected in peptides that are uniquely associated with specific regions of the protein (*Figure 3D* and *Figure 3—figure supplements 1–6*) to determine which region folds first.

These data show that the last common ancestor of ecRNH* and ttRNH*, Anc1*, as well as all proteins along the thermophilic lineage (Anc2* and Anc3*) show similar behavior to ttRNH* and gain protection first in Helix D/Strand 5 (*Figure 3D*, *Figure 3—figure supplements 1* and *2*). For the first two ancestors along the mesophilic lineage (AncA* and AncB*), the order of protection is difficult to determine. For AncA*, there is no significant difference in the degree of protection among the peptides within $I_{core}$ (this analysis is limited by the availability of peptides associated exclusively within a region) (*Figure 3—figure supplement 3*). However, when all overlapping peptides are analyzed using HDSite to obtain site resolution, we observe notable protection in Helix D at the earliest refolding times. Therefore, we conclude that although Helix D folding before Helix A is likely, the early folding events of AncA* cannot be unambiguously determined. For AncB*, all of $I_{core}$ gains protection at the same time point, both at the peptide and residue-level, so the order of assembly cannot be determined with our time resolution (*Figure 3—figure supplement 4*).

The next ancestor along the mesophilic lineage, AncC*, shows protection first in Helix D, indicating that this pattern of protection is maintained through the mesophilic lineage to this ancestor

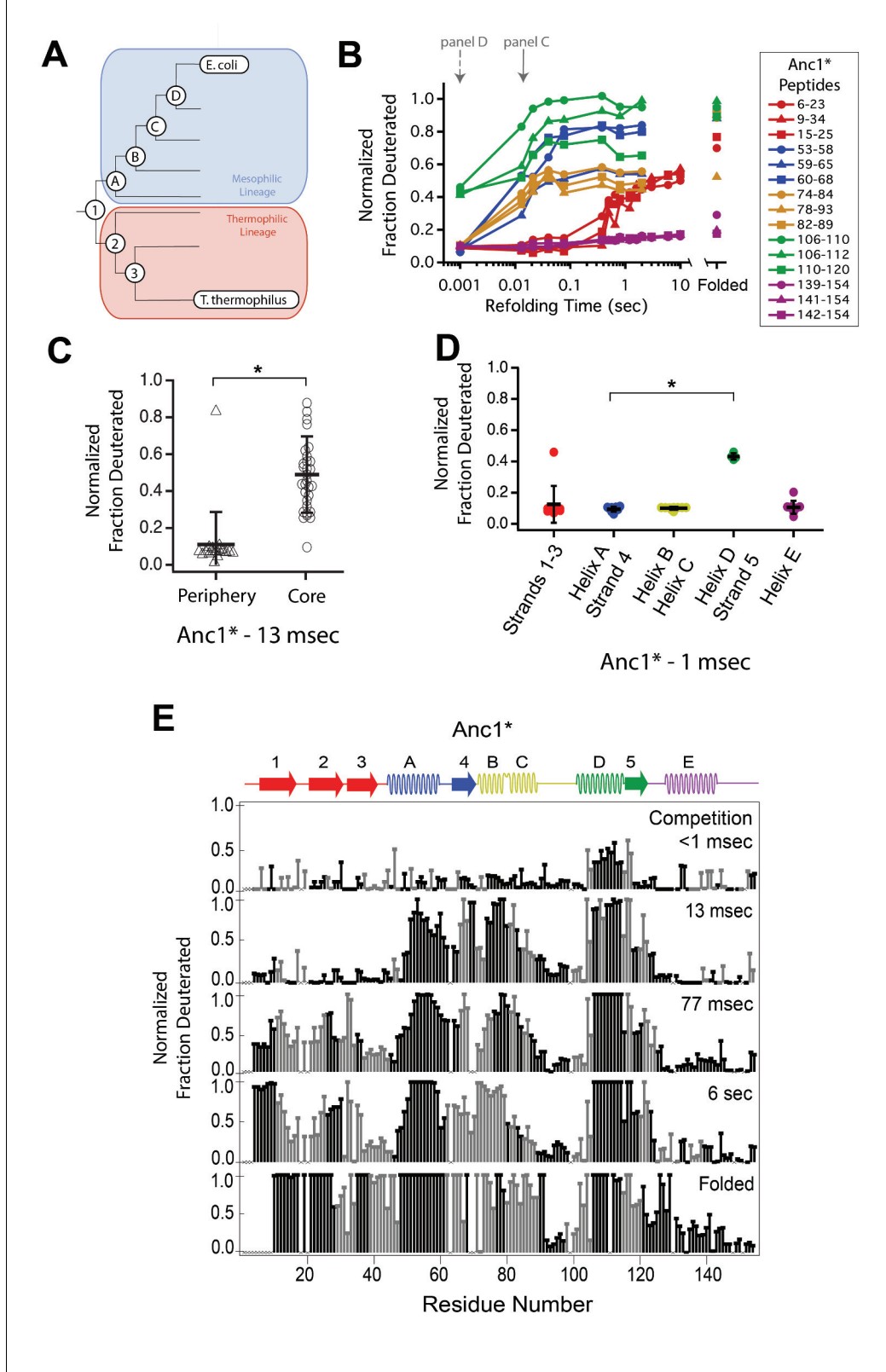

**Figure 3.** Determination of the folding pathway of ancestral RNases H by HX-MS. (**A**) Representation of the phylogenetic tree of the RNase H family illustrating the ancestral proteins along the two lineages leading to *E. coli* RNase H and *T. thermophilus* RNase H. Adapted from Figure 2A of Hart KM et al. 2014, *PLoS Biology*. 12(11) doi:10.1371/journal.pbio.1001994, published under the CreativeCommons Attribution 4.0 International Public License (CC BY 4.0; https://creativecommons.org/licenses/by/4.0/). (**Hart et al., 2014**) Anc1* is the last common ancestor of ecRNH* and ttRNH*. Anc2* and

*Figure 3 continued on next page*

*Figure 3 continued*

Anc3* are ancestors along the thermophilic lineage to ttRNH*. AncA*, AncB*, AncC*, and AncD* are ancestors along the mesophilic lineage to ecRNH*. (B) Protection of representative peptides from Anc1* at various refolding times. Peptides are colored according to their corresponding structural element. The solid arrow indicates the refolding time point analyzed in panel C. The dotted arrow indicates the refolding time point analyzed in panel D. (C) Protection of peptides mapping to the core region (I$_{core}$) or the periphery region of Anc1* at 13 msec after refolding. Bars represent the mean and standard deviation of each data set. *p = 0.0011 (Welch's unpaired T-test) (D) Protection of peptides mapping to distinct secondary structural elements of Anc1* at 1 milliseconds after refolding. Bars represent the mean and standard deviation of each data set. *p < 0.0001 (Welch's unpaired T-test). Data for C & D represent aggregate data from three separate experiments. (E) Residue-resolved folding pathway of Anc1* at representative refolding time points. Data points in black indicate residues that are site-resolved. Data points in grey indicate residues in regions with less peptide coverage and are thus not site-resolved with the neighboring residues. Residues where site-resolved protection could not be determined due to insufficient peptide coverage is denoted with a "x".

DOI: https://doi.org/10.7554/eLife.38369.007

The following source data and figure supplements are available for figure 3:

**Source data 1.** HX-MS data for ancestral RNases H.
DOI: https://doi.org/10.7554/eLife.38369.014
**Figure supplement 1.** Determination of the folding pathway of Anc2* by HX-MS.
DOI: https://doi.org/10.7554/eLife.38369.008
**Figure supplement 2.** Determination of the folding pathway of Anc3* by HX-MS.
DOI: https://doi.org/10.7554/eLife.38369.009
**Figure supplement 3.** Determination of the folding pathway of AncA* by HX-MS.
DOI: https://doi.org/10.7554/eLife.38369.010
**Figure supplement 4.** Determination of the folding pathway of AncB* by HX-MS.
DOI: https://doi.org/10.7554/eLife.38369.011
**Figure supplement 5.** Determination of the folding pathway of AncC* by HX-MS.
DOI: https://doi.org/10.7554/eLife.38369.012
**Figure supplement 6.** Determination of the folding pathway of AncD* by HX-MS.
DOI: https://doi.org/10.7554/eLife.38369.013

(*Figure 3—figure supplement 5*). AncD*, the most recent ancestor along the mesophilic lineage, however, is similar to ecRNH* and gains protection first in Helix A (*Figure 3—figure supplement 6*). As detailed for the other ancestors, the data were also analyzed using HDSite to determine residue-level protection for each ancestral RNase H (*Figure 3E*, *Figure 3—figure supplements 1–6*). These data indicate a pattern in the order of protection in the early steps of the folding pathway across the RNase H ancestors. Early protection in Helix D is an ancestral feature of RNase H that is maintained in the thermophilic lineage, with a transition occurring late during the mesophilic lineage to a different pathway where Helix A is protected before Helix D, resulting in a distinct folding pathway for the two extant RNase H homologs (*Figure 4*).

## Early helix protection is determined by the local sequence of the core

Relative to the vast sequence space available, these RNase H ancestors represent a set of closely related sequences with distinct folding properties and provide an excellent system to help us elucidate the physiochemical mechanism and the sequence determinants dictating the RNase H folding trajectory. An analysis of the intrinsic helical propensity of each region using the algorithm AGADIR (*Muñoz and Serrano, 1994*) shows a notable trend in helicity that correlates with the early folding events (*Figure 4*). For proteins that gain protection in Helix A first, the intrinsic helicity of Helix A is about four-fold higher than that of Helix D. For the variants where Helix D is protected first, the intrinsic helicity of Helix D is similar to or greater than Helix A. This suggests that intrinsic helix propensity may play an important role in determining which region is the first to gain protection during the folding pathway of RNase H. To investigate this hypothesis, we turned to rationally designed variants.

## Intrinsic helicity plays a role in determining the structure of the early intermediates

If the order of protection in the early folding events of RNase H is determined by intrinsic helix propensity, then we should be able to alter the protein sequence rationally and manipulate the folding

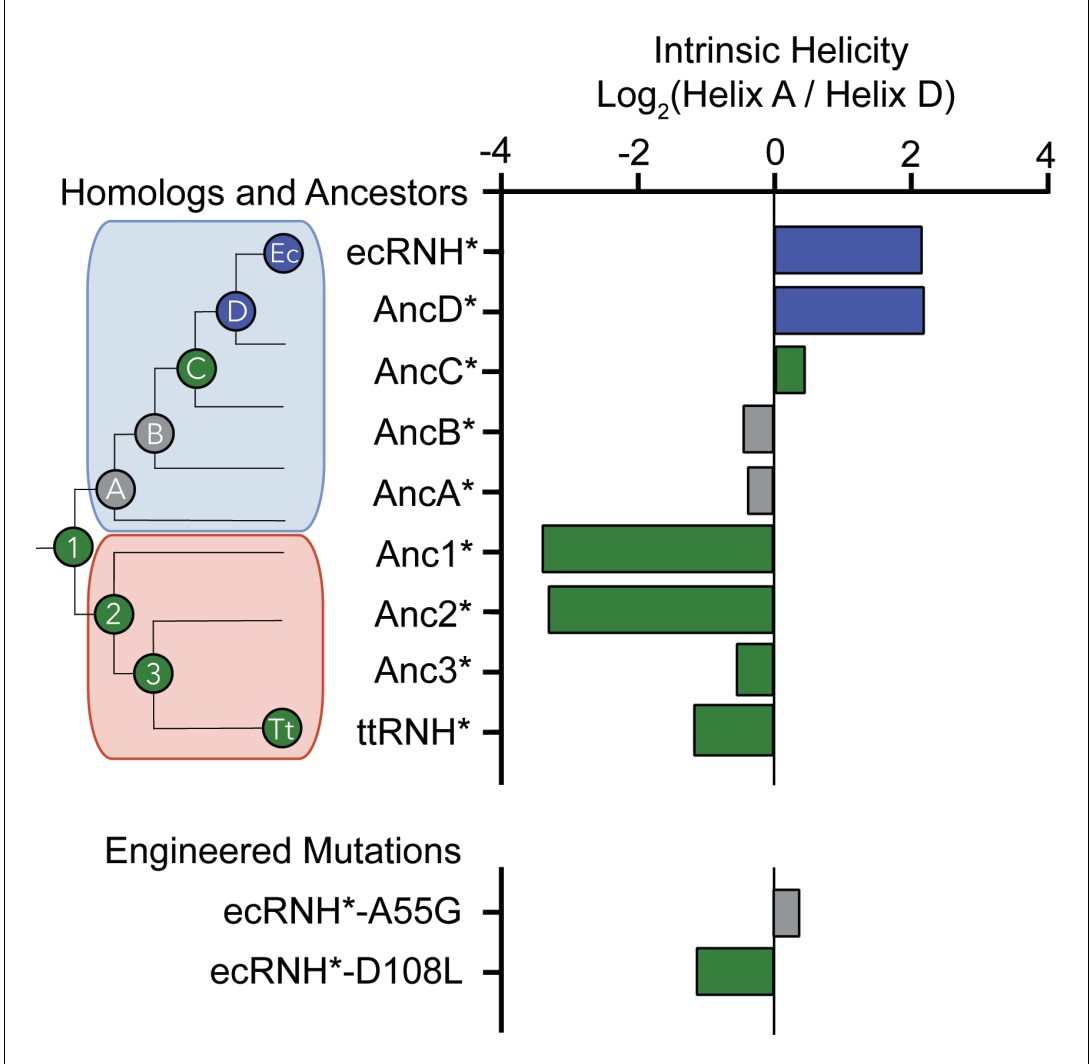

**Figure 4.** Intrinsic helicity as a predictor for the early folding mechanism of RNases H. Log-ratio of intrinsic helicity of Helix A and Helix D for each RNase H variant studied. Intrinsic helix predictions were calculated using AGADIR. (*Muñoz and Serrano, 1994*) The order of helix protection for each variant of RNase H is depicted in color. Green bars represent proteins where Helix D is the first structural element to gain protection during refolding. Blue bars represent proteins where Helix A is the first structural element to gain protection during refolding. Grey bars represent proteins where the helix protection order could not be unambiguously determined. The order of helix protection for each ancestor and homolog is also colored on the phylogenetic tree, revealing a trend in the RNase H folding trajectory along the evolutionary lineages. The phylogenetic tree shown in this figure is adapted from Figure 2A of Hart KM et al. 2014, *PLoS Biology*. 12(11) doi:10.1371/journal.pbio.1001994, published under the CreativeCommons Attribution 4.0 International Public License (CC BY 4.0; https://creativecommons.org/licenses/by/4.0/) (*Hart et al., 2014*).
DOI: https://doi.org/10.7554/eLife.38369.015

trajectory. Thus, we asked whether single-site mutations that change the relative helix propensity of Helix A and Helix D could alter the folding trajectory of ecRNH* and make it fold in a similar fashion to ttRNH*. Two different point mutations were made in ecRNH*: A55G decreases helix propensity in Helix A, and D108L increases helicity in Helix D (*Figure 4*, *Figure 5A*, *Supplementary file 1* – Table 1). Pulsed-labeling HX-MS indicates that both of these variants alter the early folding events of ecRNH*. The peptide-level protection of ecRNH* A55G indicates that at 13 msec, both Helix A and Helix D show similar levels of protection. In contrast, for wild-type ecRNH*, Helix A shows protection by 1 msec and Helix D does not show comparable protection until 10 – 20 msec (*Hu et al., 2013*). Thus, the mutation A55G slows the gain of protection in Helix A such that it no longer protected before Helix D (*Figure 5B*, *Figure 5—figure supplement 1*). The peptide-level protection of ecRNH* D108L indicates a change in the order of protection. Due to the limited number of peptides

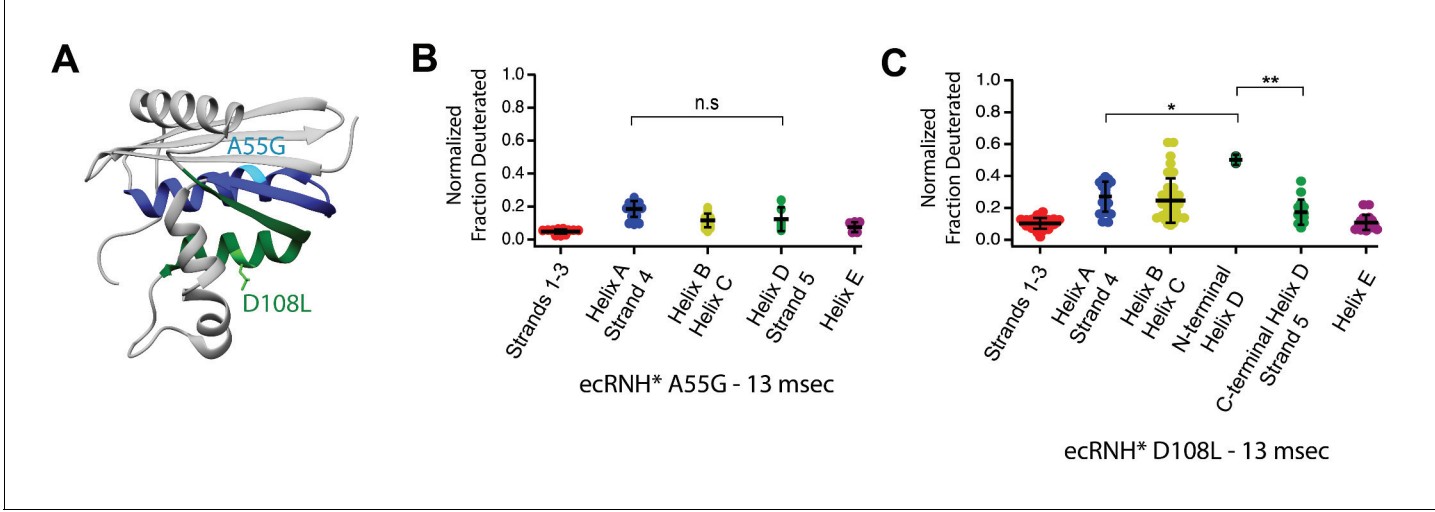

**Figure 5.** Engineered mutations to alter the folding pathway of ecRNH*. (**A**) Crystal structure of *E. coli* RNase H (PDB: 2RN2) with mutations designed to alter intrinsic helicity (**Katayanagi et al., 1992**). A55G, located in Helix A (blue), is colored in cyan. D108L, located in Helix D (green), is colored in light green. (**B**) Protection of peptides mapping to distinct secondary structural elements of ecRNH* A55G at 13 msec after refolding. Bars represent the mean and standard deviation of each data set. p=0.0917 (n.s. = not significant, Welch's unpaired T-test). (**C**) Protection of peptides mapping to distinct secondary structural elements of ecRNH* D108L at 13 msec after refolding. Bars represent the mean and standard deviation of each data set. *p=0.0044, **p=0.0016 (Welch's unpaired T-test). Data for B and C represent aggregate data from three separate experiments.

DOI: https://doi.org/10.7554/eLife.38369.016

The following source data and figure supplement are available for figure 5:

**Source data 1.** HX-MS data for ecRNH* A55G and ecRNH* D108L.
DOI: https://doi.org/10.7554/eLife.38369.018
**Figure supplement 1.** Determination of the folding pathway of ecRNH* by HX-MS.
DOI: https://doi.org/10.7554/eLife.38369.017

available, we could only confidently determine this using peptides spanning the N-terminus of Helix D. At 13 msec, the N-terminus of Helix D (residues 106 – 108) near the D108L mutation is protected significantly faster than any other region of the protein. Thus increasing helix propensity correlated with a change in the folding trajectory. (**Figure 5C**, **Figure 5—figure supplement 1**). Together, these two mutations suggest that intrinsic helicity plays a role in the early folding events of RNase H and can be used to alter the stepwise order of conformations populated during folding.

## Discussion

### Determining the folding pathway of multiple protein variants

Pulsed-labeling hydrogen exchange is currently the most detailed method to identify the conformations populated during protein folding. This approach was initially developed for use with NMR detection where it benefited from NMR's site-specific resolution of individual amides (**Bai, 2006**). However, using NMR with pulsed-labeling HX requires tens of milligrams of sample and NMR peak assignments for the amides in each protein studied. In addition, probes are limited to amide sites stable to exchange in the final folded state (protection factors of >~80,000) resulting in loss of information at individual sites, which can sometimes represent large regions of the protein. In contrast, detection by mass spectrometry as applied in this study requires much less protein sample, has much faster data collection, and can theoretically cover 100% of the protein sequence. Importantly, this approach does not demand any structural information of the folded state, such as NMR assignments, for the specific protein or variant studied. These advantages enabled us to obtain the stepwise folding pathway of nine variants of RNase H and study the evolutionary history and sequence determinants of the RNase H folding pathway in detail. While pulsed-labeling HX-MS has been used to characterize the folding pathways of several model systems, this study is the first to utilize the

higher throughput nature of HX-MS to study an ensemble of protein variants. The advantages of this technique to study many different sequences of the same fold shows great promise for probing the relationship between amino acid sequence and a protein's energy landscape and will likely be particularly valuable for protein engineering and design applications. Additionally, HX-MS should be also able to identify non-native states, backtracking, and misfolded conformations. We did not, however, observe such phenomena with our RNase H proteins.

Although pulsed-labeling HX-MS is a powerful technique, it is not without limitations. In particular, HX is a readout of the accessibility of the backbone amide to exchange with solvent, so we would not be able to distinguish between different protein conformations with the same protection pattern. Additionally, a variety of factors involved in protein folding from hydrogen bond formation and hydrophobic collapse can contribute to amide protection. Since the protection patterns we observe during folding are all consistent with protected regions in the native state, we see no evidence for non-native structure, indicating a sequential acquisition of native secondary structure.

## $I_{core}$ is a structurally conserved folding intermediate over 3 billion years of evolution

The native fold of a protein is robust to changes in sequence, and proteins with >~30% sequence identity share the same fold (*Sander and Schneider, 1991*). Thus small variations in sequence, such as those found among homologs or site-specific mutations, do not affect the overall three-dimensional structure of a protein. These mutations can, however, affect the overall energy landscape, which in turn can have profound effects on function. Here, we find conservation of a high-energy structure populated during the folding of the RNase H family over incredibly long evolutionary timescales. Using pulsed-labeling HX-MS we identified and characterized the structure of the major folding intermediate in seven ancestral and several mutant RNases H, which together with previous studies on extant homologs, suggest that the conservation of this intermediate is a key feature of the RNase H energy landscape across ~3 billion years of evolutionary time.

Why does $I_{core}$ persist on the energy landscape of RNase H? One explanation is a simple topological constraint; all RNases H may need to fold via a populated $I_{core}$ intermediate to successfully reach the native state. This explanation, however, is countered by a previous study where a single mutation (I53D) in ecRNH* destabilizes $I_{core}$ such that it is no longer populated during folding—yet this variant still folds to the native state (*Spudich et al., 2004*). Adding osmolytes, such as sodium sulfate, stabilizes this folding intermediate and switches ecRNH* I53D back to a three-state folding pathway, showing that the presence of the folding intermediate can be modulated. Additionally, a fragment of RNase H containing only the $I_{core}$ sequence (and variants thereof) can autonomously fold and be studied at equilibrium, indicating that this structure is stable and robust to mutations (*Chamberlain et al., 1999*; *Rosen and Marqusee, 2015*). The nature of the rate-limiting step, or folding barrier, which allows for the buildup of this intermediate is unclear. One possibility is that the $I_{core}$ intermediate is populated simply because the information for folding this region is completely encoded locally and $I_{core}$ can fold relatively fast, before this rate-limiting step to the fully folded state. Alternatively, the rapid collapse of $I_{core}$ may be the result of an evolutionary pressure to quickly sequester regions of the protein that might be particularly prone to aggregation. Indeed, there are several segments of RNase H that are predicted to be aggregation-prone, and most of these segments are found in the core region of the protein (*Supplementary file 1* – Table 2).

$I_{core}$ could also be conserved because it contributes to the biological function or fitness of the protein. Partially folded states and high-energy non-native conformations are known to be important for a variety of protein functions and proteostasis (*Chiti and Dobson, 2017*; *Baldwin and Kay, 2009*; *Boehr et al., 2006*). All of the ancestral RNases H we studied here are active, in that they cleave RNA-DNA hybrids in vitro (*Hart et al., 2014*); and although the residues thought to contribute to substrate-binding affinity are contained in the core region of the protein, (*Kanaya et al., 1991*) the active site residues (D10, E48, D70) span both the core and the periphery. It is therefore possible that a stable folding core with an energetically independent periphery is important for the efficiency or dynamics associated with catalysis in RNase H.

While the presence of the $I_{core}$ intermediate has been observed in all proteins studied here, recent studies have suggested that some of the RNase H variants, notably for proteins along the thermophilic lineage, the $I_{core}$ folding intermediate may also involve structure in the first β-strand (*Lim and Marqusee, 2017*; *Rosen and Marqusee, 2015*; *Zhou et al., 2008*). While we see slight

protection in this region for ttRNH\*, hydrogen exchange may not be the best probe of this—docking of Strand 1 without its hydrogen-bonding partners in the rest of the β-sheet may not be reflected by backbone amide protection. Therefore, amide protection may not be observed even if Strand 1 docks early to the core. The involvement of Strand 1 in other RNase H variants studied remains unclear from this study (*Lim and Marqusee, 2017*; *Rosen and Marqusee, 2015*).

## Aspects of the folding pathway are malleable across evolutionary time

Our pulsed-labeling HX-MS results also illustrate how other features of a protein's energy landscape can be altered over evolutionary timescales. Although the $I_{core}$ intermediate is conserved across all RNases H studied, the individual folding steps leading up to $I_{core}$ differ. Anc1\*, the last common ancestor, folds through a pathway where the Helix D/Strand 5 region is the first structural element to gain protection. This ancestral feature is maintained along the thermophilic lineage to the extant ttRNH\*. Along the mesophilic branch, we observe a switch from this ancient folding pathway to one that first forms protection in Helix A/Strand 4. This suggests that while the structure of $I_{core}$ has been conserved across 3 billion years of evolution, the steps to form this intermediate are malleable over time. Since an isolated helix is unlikely show protection by HX, we expect additional hydrophobic collapse of the polypeptide to contribute to the observed protection. Nonetheless, the switch in protection between Helix A and Helix D indicates that formation of native structure nucleates in a different region of the protein across the RNase H variants studied, with a clear evolutionary trend.

Despite these trends, it remains difficult to rationalize these observations in terms of a selective evolutionary pressure or fitness implication. These very early events occur on the order of one millisecond, significantly faster than the overall folding of the protein. Furthermore, all of these RNase H proteins fold to their native state efficiently with no evidence for aggregation or misfolding. So, although partially folded states have been implicated as gateways for aggregation for some proteins, (*Chiti and Dobson, 2017*) this does not appear to be the case for RNase H. It is possible that the change in the early folding step is a result of mutations that are coupled to another feature under selection or drift. Although the actual evolutionary implication for the RNase H folding pathway may be lost in history, the trend in folding pathway across evolutionary time demonstrates that folding pathways and conformations on the energy landscape of proteins can change over time, and this system provides an excellent tool to interrogate the role sequence plays in guiding the process of protein folding.

## The folding pathway of RNase H can be altered using simple sequence changes

Our study also shows how insights from evolutionary history can contribute to our understanding of the physiochemical mechanisms dictating the protein energy landscape and how we might use that knowledge to engineer the landscape. The regions that gain protection first involve helical secondary structure elements, and their folding order correlates with isolated helical propensity of these regions predicted by AGADIR (*Muñoz and Serrano, 1994*). Proteins where protection is first observed in Helix A have higher intrinsic helicity in Helix A than in Helix D. Proteins where Helix D gains protection first exhibit higher helicity in Helix D or roughly equal helicity in both regions. This property was used to guide our site-directed mutagenesis to select variants to alter the folding trajectory of ecRNH\* in a predictive manner using intrinsic helicity as a guide.

While these results are consistent with local helicity as a determinant of the earliest folding steps, there may be other parameters that dictate the formation of these conformations. The parameter average area buried upon folding (AABUF) (*Rose et al., 1985*) which measures the average change in surface area of a residue from an unfolded state to a folded state, has been shown to correlate to the structure of the folding intermediate in apomyoglobin (*Nishimura et al., 2005*; *Nishimura et al., 2011*). Both helicity and AABUF are altered in the mutants considered in our study (*Supplementary file 1* – Table 1). Indeed, AABUF and helicity are often correlated and contributions of either parameter are difficult to disentangle. Nevertheless, our data suggest that parameters that are locally encoded in regions of a protein can be used engineer the energy landscape of a protein including its folding pathway.

We have used a combination of ASR and pulsed-labeling HX-MS to explore the conformations populated during the folding of multiple RNase H proteins, including homologs, ancestors, and

single-site variants. All RNase H proteins studied populate the same major folding intermediate, $I_{core}$, indicating that this conformation has been maintained on the energy landscape of RNase H over long evolutionary timescales (>3 billion years). This remarkable conservation of a partially folded structure on the energy landscape of RNase H is contrasted with changes in the folding pathway leading up to this structure. The early folding events preceding this intermediate (Helix A protected before Helix D or vice versa) differ between the two homologs and also shows a notable trend along the evolutionary lineages. This pattern of protection correlates with the relative helix propensity of the sequences comprising these two helices, and we use this knowledge to alter the folding pathway of ecRNH* through rationally designed mutations. Our study illustrates how the energy landscape of a protein can be altered in complex ways over evolutionary time scales, and how insights from evolutionary history can contribute to our understanding of the physiochemical mechanisms dictating the protein energy landscape.

# Materials and methods

**Key resources table**

| Reagent type (species) or resource | Designation | Source or reference | Identifiers | Additional information |
|---|---|---|---|---|
| Strain, strain background (Escherichia coli) | BL21 (DE3) Star | QB3 Macrolab | | |
| Recombinant DNA reagent | ttRNH* (plasmid) | 10.1021/ bi982684h | n/a | |
| Recombinant DNA reagent | Anc1* (plasmid) | 10.1371/ journal.pbio.1001994 | n/a | |
| Recombinant DNA reagent | Anc2* (plasmid) | 10.1371/ journal.pbio.1001994 | n/a | |
| Recombinant DNA reagent | Anc3* (plasmid) | 10.1371/ journal.pbio.1001994 | n/a | |
| Recombinant DNA reagent | AncA* (plasmid) | 10.1371/ journal.pbio.1001994 | n/a | |
| Recombinant DNA reagent | AncB* (plasmid) | 10.1371/ journal.pbio.1001994 | n/a | |
| Recombinant DNA reagent | AncC* (plasmid) | 10.1371/ journal.pbio.1001994 | n/a | |
| Recombinant DNA reagent | AncD* (plasmid) | 10.1371/ journal.pbio.1001994 | n/a | |
| Recombinant DNA reagent | ecRNH* (plasmid) | 10.1002/ pro.5560030906 | n/a | |
| Recombinant DNA reagent | ecRNH* A55G (plasmid) | this paper | n/a | A55G point mutant of ecRNH* |
| Recombinant DNA reagent | ecRNH* D108L (plasmid) | this paper | n/a | D108L point mutant of ecRNH* |
| Commercial assay or kit | QuikChange Site-Directed Mutagenesis Kit | Agilent | 200519 | |
| Commercial assay or kit | Amicon Ultra Protein Concentrator | Millipore | UFC900324 | |
| Chemical compound, drug | Glycine | JT Baker | 4059–06 | |
| Chemical compound, drug | Urea | IBI Scientific | IB72064 | |
| Chemical compound, drug | Sodium Acetate | VWR Life Sciences | 0530–1 KG | |

*Continued on next page*

*Continued*

| Reagent type (species) or resource | Designation | Source or reference | Identifiers | Additional information |
|---|---|---|---|---|
| Chemical compound, drug | Deuterium Oxide | Sigma Aldrich | 151882–100G | |
| Chemical compound, drug | Trifluoroacetic Acid | Sigma Aldrich | 302031–100 ML | |
| Chemical compound, drug | Acetonitrile | Thermo Fisher | 85188 | |
| Software, algorithm | ExMS | 10.1007/s13361-011-0236-3 | n/a | |
| Software, algorithm | HDSite | 10.1073/pnas.1315532110 | n/a | |
| Software, algorithm | HDExaminer | Sierra Analytics | n/a | |
| Software, algorithm | Illustrator | Adobe | n/a | |
| Software, algorithm | GraphPad Prism | GraphPad Software, Inc. | n/a | |
| Other | POROS 20 R2 Beads | Thermo Fisher | 1112810 | |
| Other | Microbore column | Upchurch Scientific | C-128 | |
| Other | Reverse-phase analytical column | Thermo Fisher | 72205–050565 | |
| Other | POROS 20 AL Beads | Thermo Fisher | 1602810 | |
| Peptide, recombinant protein | Pepsin (porcine) | Sigma Aldrich | P6887 | |
| Peptide, recombinant protein | Fungal Protease Type-XIII | Sigma Aldrich | P2143 | |
| Peptide, recombinant protein | Pfu polymerase | Agilent | 600353 | |

## Protein purification

Cysteine-free *T. thermophilus* RNase H and ancestral RNases H were expressed and purified as previously described (*Hart et al., 2014*; *Robic et al., 2002*; *Hollien and Marqusee, 1999*). Point mutants were generated using site-directed mutagenesis, confirmed by Sanger sequencing, and the proteins were purified as previously described (*Dabora and Marqusee, 1994*). Purity was confirmed by SDS-PAGE and mass spectrometry.

## HX-MS system

Hydrogen exchange mass spectrometry (HX-MS) experiments were carried out using a system similar to that described by Mayne et al (*Walters et al., 2012*; *Mayne et al., 2011*). Briefly, a Bio-Logic SFM-4/Q quench flow mixer with a modified head piece with reduced swept volume was used to initiate protein refolding, followed by pulse-labeling unprotected amide hydrogen atoms, and quenching of the labeling reaction. The minimum dead time for mixing is 13 msec. Quenched samples were injected into an HPLC system constructed using two Agilent 1100 HPLC instruments. The quenched sample was flowed over columns (Upchurch C130B) packed with beads of immobilized pepsin and fungal protease at 400 μL/min in 0.05% TFA. The digested protein was run onto a C-4 trap column (Upchurch C-128 with POROS R2 beads) for desalting. An acetonitrile gradient (15 – 100% acetonitrile, 0.05% TFA at 17 μL/min) eluted peptides from this C-4 trap column and onto an analytical C-8 column (Thermo 72205–050565) for separation before injection into an ESI source for mass spectrometry analysis on a Thermo Scientific LTQ Orbitrap Discovery. The entire HPLC system is kept

submerged in an ice bath at 0°C to reduce back exchange of deuterium atoms during the chromatography steps. The workflow takes ~10–18 min from injection to peptide detection.

## Refolding experiment

Similar to previous reports, (*Hu et al., 2013*; *Mayne et al., 2011*) unfolded protein samples in high denaturant (80 µM [protein], 20 mM NaOAc pH = 4.1, 7–9 M [urea]) were deuterated by a repeated cycle of lyophilization and resuspension in $D_2O$ and full deuteration was confirmed by mass spectrometry. For the pulsed-labeling experiment, 1 volume of deuterated protein was mixed in the SFM-4/Q with 10 volumes of refolding buffer (10 mM Sodium Acetate pH = 5.29, $H_2O$) to initiate refolding. The pulse for hydrogen exchange was initiated by mixing with 5 volumes of high pH buffer (100 mM Glycine pH = 10.11) and then quenched after 10 msec with five volumes low pH buffer (200 mM Glycine pH = 1.95). The length of the delay line between the first and second mixer was changed to achieve a range of refolding times. An interrupted mixing protocol was used to measure the longest refolding time points (>373 msec). Undeuterated protein was used to perform tandem mass spectrometry (MS/MS) analysis to compile a list of peptides and their retention times in the HPLC system. Competition experiments where refolding and exchange were initiated at the same time were performed by diluting deuterated protein in high urea into high-pH refolding buffer (100 mM Glycine pH = 10.11). In this experiment each site will exchange with the solvent around it unless it can gain protection before exchange occurs (<1 msec on average). Fully folded controls were created by diluting unfolded protein samples 1:10 in fully deuterated refolding buffer and incubating at room temperature for 4 hrs before loading on the SFM-4/Q to apply the same 10 msec high-pH pulse as the other time points. All data were obtained in triplicate.

For each time point, an identical sample was collected in which the high pH pulse was omitted to measure the back-exchange (the amount of deuterium lost during the sample workup, from the quench to the injection into the mass spectrometer) for each sample. Typical back-exchange ranged from 10 – 20%, and all data except for ttRNH* were normalized to the observed back-exchange value. Data for ttRNH* were normalized to the theoretical maximum number of deuterons due to consistently poor peptide coverage in the back-exchange controls for this protein.

## MS detection and data analysis

Proteome Discoverer 2.0 (Thermo Scientific) was used to identify peptides from the tandem MS data. Peptides identified in the pulsed-labeled refolding experiments with deuterated protein were used to determine the presence and deuteration level of each peptide at each refolding time point. The spectral envelope of each peptide was fit using two separate algorithms developed by the Englander Lab to determine their deuteration state — ExMS for identification and fitting of peptides and HDsite for deconvolution of overlapping peptides to achieve near-amino acid level deuteration levels (*Kan et al., 2013*; *Kan et al., 2011*). Site-resolution was determined by peptide coverage of each protein at each time point. If adjacent residues were not site-resolved by peptide coverage but their normalized fraction deuterium value was within <0.1 of each other, these residues were also considered to be site-resolved. In addition, HDExaminer (Sierra Analytics) was used to identify and fit each peptide and determine deuteration levels. Different charge states of the same peptide were averaged where noted and used for further analysis. Centroids of each peptide at each time point taken from HDExaminer were used for further analysis. The residue cutoffs for specific structural regions of each protein were determined from a multiple sequence alignment using the structure of *E. coli* RNase H as a guide (PDB: 2RN2) (*Hart et al., 2014*). Peptides were assigned to different structural regions based on these residue cutoffs. Peptides that spanned multiple secondary structural regions of a protein were excluded from further analysis, as were peptides not present in all time points. Peptides mapping to Strands 1 – 3 and Helix E were assigned to the periphery region of the protein. Peptides mapping to Helix A-D and Strands 4 – 5 were assigned to the core region of the protein. Comparison of protection from different groups was carried out using GraphPad Prism (GraphPad Software). Peptides from all experimental replicates were aggregated, and the distributions of groups were compared pairwise using Welch's unequal variances t-test. Statistically significant differences in the mean are reported with their associated p-values throughout the text and figures where appropriate.

## Acknowledgements

We thank members of the Marqusee Lab for helpful discussions, and Dr. Ha Truong for assistance with the aggregation prediction analyses. We also thank Dr. Leland Mayne and members of the Englander Lab and Dr. Goran Stjepanovic in the Hurley Lab for support on the pulsed-labeling HX-MS instrumental setup and analysis. We thank the Vincent J Coates Proteomics/Mass Spectrometry Facility for instrumentation support. This work was funded by NIH Grant GM050945 (to SM) and a National Science Foundation Graduate Research Fellowship (to SAL). SM is a Chan Zuckerberg Biohub Investigator.

## Additional information

### Funding

| Funder | Grant reference number | Author |
|---|---|---|
| National Institute of General Medical Sciences | GM050945 | Shion An Lim<br>Eric Richard Bolin<br>Susan Marqusee |
| National Science Foundation | Graduate Research Fellowship | Shion An Lim |

The funders had no role in study design, data collection and interpretation, or the decision to submit the work for publication.

### Author contributions

Shion An Lim, Conceptualization, Data curation, Formal analysis, Investigation, Writing—original draft, Writing—review and editing; Eric Richard Bolin, Conceptualization, Data curation, Formal analysis, Investigation, Methodology, Writing—original draft, Writing—review and editing; Susan Marqusee, Conceptualization, Supervision, Funding acquisition, Project administration, Writing—review and editing

### Author ORCIDs

Shion An Lim (iD) http://orcid.org/0000-0003-2136-2732
Eric Richard Bolin (iD) http://orcid.org/0000-0002-9265-0451
Susan Marqusee (iD) http://orcid.org/0000-0001-7648-2163

### Decision letter and Author response

Decision letter https://doi.org/10.7554/eLife.38369.022
Author response https://doi.org/10.7554/eLife.38369.023

## Additional files

### Supplementary files

• Supplementary file 1. Table 1: Comparison of intrinsic helicity and AABUF across RNase H variants Values for the helicity and average area buried upon folding (AABUF) are presented for Helix A and Helix D in each of the RNase H variants studied. Values represent the average value acorss the entire helix and are calculated based on previous work (*Muñoz and Serrano, 1994*; *Rose et al., 1985*; *Nishimura et al., 2005*). Table 2: Predicted aggregation-prone segments in the RNase H sequence Aggregation-prone sequences found in the RNase H variants studied were predicted using several methods based on primary sequence (*Goldschmidt et al., 2010*; *Gasior and Kotulska, 2014*; *Zhou et al., 2008*). Sequences are listed with their location within the protein, which proteins contained the sequence, and the method used to identify them.
DOI: https://doi.org/10.7554/eLife.38369.019

• Transparent reporting form
DOI: https://doi.org/10.7554/eLife.38369.020

## Data availability

All data generated or analyzed during this study are included in the manuscript main text, supporting files, and source data.

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
