## [Decision Letter]

Thank you for submitting your manuscript "Tracing a protein's folding pathway over evolutionary time using ancestral sequence reconstruction and hydrogen exchange" for consideration. Your paper has been reviewed by three peer reviewers, one of whom is a member of our Board of Reviewing Editors, and the evaluation has been overseen by a Senior Editor. The consensus is that the work is of general interest, although some revision is requested. No further experiments are required at this time.

Summary:

The reports a landmark, high-resolution determination of structure formation during the folding process of two extant ribonuclease H homologs (from *T. thermophilus* and *E. coli*, ttRNH* and ecRNH*, respectively), a series of their reconstructed ancestral progenitors reaching back to their last common ancestor, and designed site-directed point mutations of ecRNH* that test the molecular basis for differences in folding. The biophysical analysis of this conceived set of variants provides a unique view of the molecular basis for how the energy landscape of protein folding may be modulated over the course of evolution. As there is very little such information currently known, this study provides unprecedented experimental detail to address critical, fundamental questions regarding how evolutionary variations in primary amino acid sequence determine tertiary structure. Such information has far-reaching impact, as the authors demonstrate how this powerful H/D method developed previously and applied to ecRNH* (PNAS 2013) may be widely applied and provide testable knowledge of folding that is important for advancing outstanding challenges in the accurate modelling of folding as well as in protein engineering and design. The key findings are that an obligatory partially folded intermediate is conserved over 3 billion years of RNH evolution, but the path to this intermediate can vary, both during evolution and by design. A significant modulator of the protein energy landscape is the intrinsic helical propensity of different secondary structural elements, i.e. helix A and helix D. Both A and D helices are formed in the conserved intermediate, but helix D is the first secondary structure element to fold in the common ancestor and in variants up to and including ttRNH*. In contrast, along the phylogenetic branch leading to ecRNH*, the folding of helix A is progressively increasingly favourable compared to helix D, such that helix A folds first in ecRNH and the preceding ancestral protein (AncD*). The relative formation of helix A and D during folding is mirrored in the corresponding relative intrinsic helical propensities in the different RNHs, and is demonstrated as rationally alterable by characterizing point mutants engineered to change helical propensity.

Although all reviewers enjoyed the paper and recognized the novelty of the work it was felt that it should be stressed more completely in the revised manuscript. This is particularly the case as several previous papers have been published by this group on the same RNH system, although structural information on the folding order is not provided there.

Minor points:

- Do the authors think that there is a functional significance at all to the order of helix folding (D before A or vice versa)?

- Is the sequence that folds first, corresponding to the intermediate, more aggregation prone than the slower folding remainder? Could it be that the goal of the intermediate is just to sequester quickly sticky amino acids?

- Could the authors discuss a bit about the limitations of their hydrogen exchange method in terms of generating intermediate structures. They stress the positives of course in their paper. Yet, it seems like it is not possible to distinguish between the formation of different structural elements in cases where protection would be the same and hence the requirement for a somewhat native-centric folding model.

- Additional information should be provided on analysis of the HX-MS data and on the associated uncertainties. This is important to better define the current results and for other investigators who may use the results or the HX-MS method in future. For example, what are the estimated uncertainties in the residue-resolved data (panels D or E)? What specific criteria are used to assign results for an individual residue to grey (increased uncertainty) or black? Why does the grey/black colour change for different folding timepoints of the same protein? Some additional detail on how differences in protection of secondary structural elements (panels C or D) were calculated to be statistically different is also warranted. In panel C/D (and B/C), I guess each point represents data for 1 peptide in 1 experiment, or were the data determined in triplicate averaged? Please also clarify details regarding how the various normalizations were done and the extent of back exchange and the impact of back exchange on uncertainties. What are the impacts/significance for data interpretation of normalizing the data for ttRNH* to the theoretical maximum (e.g. will fractions deuterated tend to be systematically low in Figure 2). Much of this information could be in the supplementary information, perhaps including additional specific information on what peptides were observed/analyzed for different proteins.

- The authors discuss the pertinent point that the level of protection of the peripheral secondary structural elements (strands 1-3 and Helix E), which are not protected in the intermediate, differs in the folded RNHs. To what extent this different protection may impact interpretation of the results? Also, in principle the HX-MS may reveal non-native intermediates e.g. perhaps there is more structure in helix A in the intermediate than in the folded state (e.g. for Anc3*). This may also be worth commenting on?

- Comparison of the new results described in this paper with results obtained previously for ecRNH* (PNAS 2013) are central to the conclusions of this paper, but no data are shown for ecRNH*. The ecRNH* results were obtained at 25°C rather than the 10°C used for the new HX-MS results here, and that AncD*, like ecRNH*, shows protection of Helix A before Helix D. Still, it may be helpful to include some additional information on ecRNH* both for HX-MS and for folding kinetics in Figure 2—figure supplement 1. Please clarify the significance/relationship of the 2 phases observed for ttRNH* for the HX-MS results, and whether/why 1 or 2 phases are observed for the other proteins.

- Figure 5 shows HX-MS at an early timepoint of folding for ecRNH* point mutations A55G (decreases helical propensity of Helix A) and D108L (to increase helicity of D). It would certainly be of interest to see also the corresponding residue resolved analysis, and if the observed effects persist throughout folding (i.e. add the equivalent of panel D).

---

## [Author Response]

[…] Although all reviewers enjoyed the paper and recognized the novelty of the work it was felt that it should be stressed more completely in the revised manuscript. This is particularly the case as several previous papers have been published by this group on the same RNH system, although structural information on the folding order is not provided there.

We are pleased that the reviewers share our excitement about the novelty of our work and its contribution to the field. We have added several sentences in the Introduction and Discussion section to further emphasize the novelty and key findings of the paper.

Minor points:- Do the authors think that there is a functional significance at all to the order of helix folding (D before A or vice versa)?

Because the earliest events in folding where helix D or helix A gain protection takes place very quickly (on the order of milliseconds), we found it difficult to rationalize a functional significance to this trend and have outlined our reasoning in the Discussion section (subsection “Aspects of the folding pathway are malleable across evolutionary time”). Although it is possible that intrinsic helicity can underlie functional effects in the folded state of the protein (i.e. substrate binding, native-state dynamics, catalysis) or be involved in stability (generally the more thermodynamically stable proteins show protection in helix D first), we can only speculate since we did not specifically probe the functional significance of the observed folding trends in this study.

- Is the sequence that folds first, corresponding to the intermediate, more aggregation prone than the slower folding remainder? Could it be that the goal of the intermediate is just to sequester quickly sticky amino acids?

Thank you for bringing up this interesting point. To examine this in the RNase H family, we ran three different aggregation predictors (Zipper DB, FISH Amyloid, AmylPred) on the RNase H sequences and we summarize the results in a new supplementary table (Table 2 in Supplementary file 1). The algorithms found several aggregation-prone protein segments, most of which are in the core region of the protein. Thus, as the reviewers hypothesized, it is entirely possible that the goal of the intermediate is to quickly sequester these sticky amino acids. We did not observe any aggregation-like behavior in our HX-MS experiments, but investigating whether this is a significant factor in RNase H folding and evolution would be an interesting future direction. We have also added some text in the Discussion to address this in the manuscript (subsection “I_core_ is a structurally conserved folding intermediate over 3 billion years of evolution”, second paragraph).

- Could the authors discuss a bit about the limitations of their hydrogen exchange method in terms of generating intermediate structures. They stress the positives of course in their paper. Yet, it seems like it is not possible to distinguish between the formation of different structural elements in cases where protection would be the same and hence the requirement for a somewhat native-centric folding model.

Thank you for this important feedback about the advantages and disadvantages of HX-MS. It is true that hydrogen exchange measures protection, and cannot distinguish between different structures with the same residue protection. In this manuscript, we have done our best to refrain from drawing conclusions about specific structures of the partially folded states we observe in the RNase H proteins, and instead we focus on describing our data as protection during folding that are consistent with protected regions in the native state. We have reviewed our manuscript to ensure that our data interpretation and wording is consistent with this approach. Additionally, we have added language in the main text (subsection “Determining the folding pathway of multiple protein variants”) that explains both the advantages and disadvantages of hydrogen exchange, focusing particularly on the limitations and assumptions involved in inferring partially folded structures.

- Additional information should be provided on analysis of the HX-MS data and on the associated uncertainties. This is important to better define the current results and for other investigators who may use the results or the HX-MS method in future. For example, what are the estimated uncertainties in the residue-resolved data (panels D or E)? What specific criteria are used to assign results for an individual residue to grey (increased uncertainty) or black? Why does the grey/black colour change for different folding timepoints of the same protein? Some additional detail on how differences in protection of secondary structural elements (panels C or D) were calculated to be statistically different is also warranted. In panel C/D (and B/C), I guess each point represents data for 1 peptide in 1 experiment, or were the data determined in triplicate averaged?

We have provided additional details in the Materials and methods section and figure legends to clarify how the HX-MS data were analyzed and whether the displayed data is an average, aggregated, or a representative example from replicates. We’d like to clarify that the grey/black coloring for the site-resolved data may be different for different folding time points because site-resolution was determined by the peptide coverage of each sample. For the same protein, slightly different peptides (both in identity and number) may be detected for any given MS run. We hope that this additional information can provide the details necessary for accurate interpretations of our work and for future investigators to build upon our work.

Please also clarify details regarding how the various normalizations were done and the extent of back exchange and the impact of back exchange on uncertainties. What are the impacts/significance for data interpretation of normalizing the data for ttRNH* to the theoretical maximum (e.g. will fractions deuterated tend to be systematically low in Figure 2). Much of this information could be in the supplementary information, perhaps including additional specific information on what peptides were observed/analyzed for different proteins.

We have provided additional details in the Materials and methods section to clarify how the HX-MS data was normalized and stated the extent of back-exchange observed in our experiments. As previously explained in our Materials and methods, for one protein, ttRNH*, we chose to normalize the data to the theoretical maximum because of the consistently poor peptide coverage in the back-exchange control (this may be due to its high stability and difficulty in peptide cleavage). Since the choice of normalization only affects the absolute value but not the relative differences between time points or secondary structure elements in deuterium protection, we believe the data interpretation and the statistical significance should not be affected in the ttRNH* dataset.

Additionally, we have provided supplementary information tables (Figure 2—source data 1, Figure 3—source data 1, Figure 5—source data 1) that lists the source data for each protein studied in this manuscript, the identity of all of the peptides and their normalized fractional deuterium content at each time point.

- The authors discuss the pertinent point that the level of protection of the peripheral secondary structural elements (strands 1-3 and Helix E), which are not protected in the intermediate, differs in the folded RNHs. To what extent this different protection may impact interpretation of the results? Also, in principle the HX-MS may reveal non-native intermediates e.g. perhaps there is more structure in helix A in the intermediate than in the folded state (e.g. for Anc3*). This may also be worth commenting on?

As the reviewer noted, we see interesting differences in the level of protection in the folded state across the different RNase H proteins. These data are consistent with previous results, such as the non-2-state equilibrium behavior of AncB* (Hart et al., 2014) and the relative lability of Helix E (Goedken, Raschke and Marqusee, 1997). For all of these proteins, I_core_ forms during folding and is structurally conserved, and the folding steps to I_core_ take place on a timescale much faster than typical folding of the periphery (msec vs. sec). Thus, we believe that differences in the native ensemble of the protein would not impact the interpretation of the results and the trends we observe regarding the early folding steps leading up to the conserved I_core_ intermediate.

As the reviewer noted, it should be possible to reveal non-native intermediates with HX-MS, which we believe is a very important and powerful application of this technique. Specifically to Anc3*, however, we would like to clarify to the reviewer that the residue-resolved data spanning Helix A (Figure 3—figure supplement 2D) is missing because of a lack of peptide coverage in that region for that time point. When site-resolved data is not available, we use the symbol “x” to denote the lack of data at those residues, which the reviewer may have inadvertently interpreted as low protection values.

- Comparison of the new results described in this paper with results obtained previously for ecRNH* (PNAS 2013) are central to the conclusions of this paper, but no data are shown for ecRNH*. The ecRNH* results were obtained at 25°C rather than the 10°C used for the new HX-MS results here, and that AncD*, like ecRNH*, shows protection of Helix A before Helix D. Still, it may be helpful to include some additional information on ecRNH* both for HX-MS and for folding kinetics in Figure 2—figure supplement 1.

First, thank you for pointing out that the manuscript would be more complete as a stand alone entity if we provided more information on ecRNH* for comparison. We have added this to the manuscript (see below). However, we would like to clarify an important point – the ecRNH* HX-MS data in the Hu et al., 2013 paper was actually conducted at 10°C (and not 25°C as suggestion by the reviewers), and is therefore under exactly the same conditions as the results in this manuscript. To help clarify this and provide a complete set of data on all the discussed proteins in this manuscript, we have included an additional panel (panel K) in Figure 2—figure supplement 1 showing the chevron plot for ecRNH* at both 25°C and 10°C. The data shown are identical to Figure 1C in the Hu et al., 2013 manuscript and re-plotted to be consistent with the other panels in Figure 2—figure supplement 1. Additionally, we have provided the HX-MS results for ecRNH* from the Hu et al., 2013 manuscript and presented it as a new supplementary figure (Figure 2—figure supplement 2).

Please clarify the significance/relationship of the 2 phases observed for ttRNH* for the HX-MS results, and whether/why 1 or 2 phases are observed for the other proteins.

Except for ttRNH*, all of the other RNases H studied here show a single observable phase and a burst phase (see Lim et al., 2016). It has been known for some time that the observable refolding kinetics for ttRNH* fits best to a biphasic exponential process (by CD spectroscopy). The underlying mechanism and structural implications of this second phase were studied extensively in a previous manuscript (Hollien and Marqusee, 2002). Although there is some evidence that proline isomerization is involved, the exact molecular process responsible for the second phase of ttRNH*, which was not observed in any other ancestral RNases H (see Lim et al., 2016), remains elusive. We do not see any evidence of a second phase in the pulse labeling HX-MS data. It would have been very exciting if our HX-MS data revealed two distinct phases in the rate-limiting step unique to ttRNH*. We have added a section in the manuscript (subsection “Monitoring the folding pathway of ttRNH* using pulsed-labeling HX-MS”, first paragraph) that expands on this topic, and acknowledge that this remains an open question in RNase H folding, which we hope a future HX-MS experiment or another method could eventually uncover.

- Figure 5 shows HX-MS at an early timepoint of folding for ecRNH* point mutations A55G (decreases helical propensity of Helix A) and D108L (to increase helicity of D). It would certainly be of interest to see also the corresponding residue resolved analysis, and if the observed effects persist throughout folding (i.e. add the equivalent of panel D).

As suggested, we have added a supplementary figure (Figure 5—figure supplement 1) that shows the residue-resolved data for ecRNH*-A55G and ecRNH*-D108L. For these two proteins, we only carried out the earliest folding time points and the folded state control in order to address the question and hypothesis of which helix is protected first.